# Identifying primary-care features associated with complex mental health difficulties

Ciarán D. McInerney[1]*, Phillip Oliver[1], Ada Achinanya[1], Michelle Horspool[2], Vyv Huddy[3], Christopher Burton[4]

1 School of Medicine & Population Health, University of Sheffield, Sheffield, England, 2 Sheffield Health and Social Care NHS Foundation Trust, Sheffield, United Kingdom, 3 School of Psychology, University of Sheffield, Sheffield, England, 4 Sheffield Centre for Health & Related Research, University of Sheffield, Sheffield, England

* ciaran.mcinerney@sheffield.ac.uk

## Abstract

### Aim

The coded prevalence of complex mental health difficulties in electronic health records, such as personality disorder and dysthymia,is much lower than expected from population surveys. We aimed to identify features in primary care records that might be useful in promoting greater recognition of complex mental health difficulties.

### Methods and Findings

We analysed Connected Bradford, an anonymised primary care database of approximately 1.15M citizens. We used multiple approaches to generate a large set of features representing multi-level collections of patient attributes across time and dimensions of healthcare. Feature sets included antecedent and concurrent problems (psychiatric, social and medical), patterns of prescription and service use and temporal stability of attendance. These were tested individually and in combination. We analysed the relationship between features and diagnostic codes using scaled mutual information.

We identified 3,040 records satisfying our definition of complex mental health difficulties. This was 0.3% of the population compared to an expected prevalence of 3–5%. We generated >500,000 features. The most informative feature was count of unique psychiatric diagnoses. Other features were identified, including binary features (e.g., presence or absence of prescription for antipsychotic medication), continuous features (e.g., entropy of non-attendance) and counts of features (e.g., concerning behaviours such as self-harm & substance misuse). Several of these showed odds ratios >=5 or <=0.2 but low positive predictive value. We suggest this is due to the large number of "cases" being uncoded and, thus appearing as "controls".

**Data availability statement:** This study is based on data from Connected Bradford (NHS REC 22/EM/0127), which can only be accessed by approved users of the Connected Bradford data platform. The data contain sensitive patient information and are owned by a third-party. The Connected Bradford Research Database is hosted by Bradford Teaching Hospitals NHS Foundation Trust and cannot be shared publicly. Data can only be made accessible to researchers upon completion of a data access application and is submitted to the Connected Bradford Governance Board for scientific review. If the application is approved, then researchers are provided access to a virtual environment to undergo the analysis. All outputs are reviewed by the Board before dissemination. For further information on the data access process, please contact Connected Bradford on cBradford@bthft.nhs.uk or via this site https://www.bradfordresearch.nhs.uk/our-research-teams/connected-bradford/".

**Funding:** Funded by the National Institute for Health Research Research for Patient Benefit - "Mental Health in the North" call (NIHR203473) the funders had no role in study design, data collection and analysis, decision to publish, or preparation of the manuscript.

**Competing interests:** The authors have declared that no competing interests exist.

## Conclusion

Complex mental health difficulties are poorly coded. We demonstrated the feasibility of using information theoretic approaches to develop a large set of novel features in electronic health records. While these are currently insufficient for diagnosis, several can act as prompts to consider further diagnostic assessment.

## Introduction

Complex mental health difficulties is a generic term to describe difficulties more persistent or disruptive than the common mental disorders but which do not meet current definitions of severe mental illness like psychosis or bipolar disorder [1,2]. They are characterised by repeated episodes of anxiety and depression, with long-term unpredictable changes in mood and difficulties in relationships. This means there are overlaps with diagnostic entities including personality disorders, persistent depression (dysthymia), co-morbid substance misuse, neurodevelopmental issues and the consequences of trauma. Although complex mental health difficulties overlap with these diagnoses [3], it is the interaction of fluctuating presentation, comorbidities, social context, treatments and support needs that indicate complex mental health difficulties [2] rather than any diagnosis per se.

The nature of complex mental health difficulties and the design of healthcare systems can make it difficult to provide satisfactory care [4–6]. Care is often episodic and crisis-related, and is delivered in general practice and emergency departments [7]. This fractured care challenges the development of constructive working alliances between patients and healthcare professionals, which are essential for continuity of care, and diagnostic clarity [8–10].

Complex mental health difficulties are common. The global prevalence of personality disorder is estimated to be 6% of adults (standard error = 0.3%), and approximately 4% (95% confidence interval: 2.9–6.7) in the UK [11,12]. Two studies have examined the prevalence of personality disorder in primary care patients attending with mental health difficulties and found rates of 23.8% and 25.5% in UK and Finish (mental health clinics) settings, respectively [13,14]. However, rates of diagnostic coding are much lower for disorders like personality disorder, in primary-care electronic healthcare records. A UK EHR database study found only 1.28% of patients in a national database had a diagnostic code for personality disorder (assuming an estimated sampled population of 3.6 million); a Catalonian study reporting only 0.017% for borderline personality disorder; and a Norwegian study reported 0.89% incidence within a one-year period [15–17]. While depression can be recognised from codes or prescriptions in EHRs. Similarly-low primary care coding rates have been observed for post traumatic disorders and persistent depression [18]. It is not clear if these low levels of coding represent under-recognition (perhaps due to practitioners working to guidelines for common mental disorders, like depression and anxiety), recognition without diagnosis (for instance because of limited access to specialist care), or under-coding of established diagnoses (because some

diagnoses are seen as stigmatising). Whichever the cause, identifying people with diagnoses indicative of complex mental health difficulties is important because some evidence suggests they respond less well to treatments established for common mental disorders [19,20] and their unmet care needs are overlooked [21]. There are precedents of identifying undiagnosed patients in several fields of healthcare [22–25]. For example, a study attempted to identify people with personality disorder from a US dataset involving mental healthcare and emergency department visits [25]. We were unable to find similar approaches using a primary care dataset.

We conducted a mixed-methods study entitled Understanding Services for people with Complex Mental Health Difficulties (UnSeen) [26]. A qualitative component examined the ways that patients and general practitioners conceptualise complex mental health difficulties and how this relates to primary care [26]. The quantitative component, described here, analysed a large, anonymised dataset of electronic healthcare records to ask the question *What features within patients' primary-care electronic healthcare records might indicate that a patient has complex mental health difficulties?*

Our study is the first to look for signals of complex mental health difficulties in primary-care electronic health records. We are also one of the first to use an information-theoretic approach [27] and introduce the concept of features to represent multi-level collections of patient attributes across time and dimensions of healthcare. Specifically, we identified sets of features based on their two-way mutual information with a variable defining the caseness of complex mental health difficulties. This "mutual information" statistic is a scaled measure of coincidence of two variables, comparing the assumption of statistical dependence with the assumption of statistical independence [28] (note that in the original source [28], Robert Fano used the phrase "expectation of the mutual information" to refer to what we now simply call mutual information). We use an information-based measure rather than a regression or classification measure because it makes the fewer assumptions about the form of the relationship between variables, which are unknown to us at this exploratory stage (e.g., directionality, linearity, distribution of residuals, relative weighting of true positives and true negatives). Also, information theory is suitable for medical decision making and the study of healthcare records because the notion of coincidence (rather than probability) lends itself to the binary nature of diagnoses and records thereof. The remainder of this manuscript describes our methods, presents the feature sets we identified that can help healthcare services to provide appropriate and timely care, and discusses what might be needed to improve the identification of complex mental health difficulties.

## Materials and methods

This is a retrospective study of electronic healthcare records using a case-control design. Data were accessed between 19th July 2022 and 30th November 2023. No author had access to information that could identify individual participants. We used an information-theoretic approach to identify features within patients' primary-care electronic healthcare records that were associated with our provisional definition of complex mental health difficulties. We used a mixture of R (4.2.3), Python (3.7.12), and Google Big Query (2.0.96). All scripts are available in the study's GitHub repository at https://github.com/ConnectedBradford/CB_1759_Joining-Primary-and-Secondary-Care-. Specific notebooks from this repository are referenced throughout this manuscript.

### Study population and data source

Connected Bradford is a health research database connecting de-identified, longitudinal, near-real-time data from different organisations across the Bradford and Airedale region of England, UK [29]. It uses a whole-system framework that links patient's electronic healthcare records with regional databases covering housing, social welfare, crime, education, environment, and more [30]. Consent to use healthcare data for research is granted via the UK National data opt-out scheme prior to acquisition by Connected Bradford [31]. At the time of this study, the primary-care database excluded "sensitive" clinical codes (supplementary material S1 Table).

The study population is intended to represent patients within the Connected Bradford database who have or might have complex mental health difficulties. As such, it comprises anyone with mental ill-health but who does not have a diagnosis of a severe mental illness (specifically, schizophrenia or bipolar disorder). This was defined as: any person with a record in the primary-care tables within the Connected Bradford database; aged between 18 and 70, inclusive; who have been registered with their general practice for at least one year; who, additionally, either have a record of a SNOMED-CT diagnostic code of interest within ten years prior to 31ˢᵗ December 2021 (see link associated with "Mental disorder | SCTID: 74732009 + child codes" in S2 Table), or who have a record of prescriptions for medicines of interest (Table 1) within ten years prior to 31ˢᵗ December 2021; excluding those people who have a record of a SNOMED-CT diagnostic code of schizophrenia or bipolar disorder.

Our SNOMED-CT diagnostic codes of interest were defined by the complete list of SNOMED-CT concept descendants of '74732009 | Mental Disorder' within the Clinical Finding domain. SNOMED-CT codes for schizophrenia and bipolar were excluded because these conditions indicate severe mental ill-health that is outside the scope of this study. Codes for dementia were excluded because they might warrant off-label prescribing of antipsychotics despite not being associated with personality disorder. Our medicines of interest were specified by the General Practitioner members of the research team to represent commonly-used antidepressants, anxiolytics, hypnotics, and antipsychotics. All codelists used in this study can be found by following the URLs in supplementary material S2 Table.

## Data governance and management

Access to Connected Bradford database was via Google Cloud Platform. Bradford Teaching Hospitals NHS Foundation Trust is the data controller of the Connected Bradford database. The project Principal Investigators (co-authors PO and CB) hold overall responsibility for data management.

Institutional ethical approval was granted for our particular study (University of Sheffield; ref 047008), and for all research using the Connected Bradford data (NHS HRA reference: 22/EM/0127). All data were fully anonymized before access.

**Table 1. Parent SNOMED-CT clinical codes used to define the caseness of complex mental health difficulties. All child codes of parent codes were searched, too. Also showing medication names used to indicate active caseness (i.e., ten years prior to the 31ˢᵗ December 2021). Medications were searched by name. The full codelist is available in supplementary material S2 Table.**

|  | Diagnosis (Parent and child SNOMED-CT code) |
|---|---|
| Inclusion | Borderline personality disorder (20010003) |
|  | Chronic depression (192080009) |
|  | Chronic post-traumatic stress disorder (313182004) |
|  | Complex posttraumatic stress disorder (443919007) |
|  | Dysthymia (78667006) |
|  | Personality disorder (33449004) |
|  | Persistent depressive disorder (1153575004) |
|  |  |
| Exclusion | Bipolar disorder (13746004) |
|  | Dementia (Various) |
|  | Schizophrenia (58214004) |
|  | **Medication** |
| Antidepressants | Clomipramine; Citalopram; Duloxetine; Escitalopram; Mirtazapine; Paroxetine; Sertraline; Trazodone; Venlafaxine; Fluoxetine |
| Antipsychotics | Risperidone; Olanzapine; Quetiapine; Flupentixol; Chlorpromazine; Aripiprazole; Haloperiodol |
| Hypnotics & Anxiolytics | Diazepam; Zopiclone |

Disclosure refers to the re-identification of individuals, households, or organisations, should data users attempt to do so [32]. To produce descriptive statistics and feature sets, raw counts were redacted if less than or equal to seven, then rounded to the nearest ten, before any calculations.

## Caseness variable definition

We distinguished cases and controls within the records of our study population. 'Cases' were those records in our study population that demonstrate recent, active, complex mental health difficulties, defined by having 1) at least one SNOMED-CT diagnostic code from our inclusion list, excluding those from our exclusion list from any time prior to 31st December 2021, and 2) a prescription for medicines of interest within ten years prior to 31st December 2021 (Table 1). 'Controls' were those records that did not meet at least one of these criteria. We chose our index date of 31st December 2021 because it was the date before which prescription data had been reliably updated, at the time of analysis.

The requirement for medication within the specified period was to ensure that caseness was only met if an individual had recently been prescribed medication for their mental health. This was motivated by preliminary searches in the clinical team members' own practices that had identified a small number of individuals with a single code for personality disorder years before evidence of current mental ill-health.

We accept that there will have been some contamination across groups because our definitions were designed to separate caseness and control-ness with the purpose of screening patients who might have unmet needs (Fig 1). We have attempted to maximise the specificity of our caseness definition in expectation that cases will be a small portion of the population [15–17].

## Features and feature sets

Our aim was to identify features within patients' primary-care electronic healthcare records that might indicate that a patient as has complex mental health difficulties. We defined features as any patient attribute available from patients' primary-care electronic health record that is not the definition of caseness variable. At their simplest, features were binary indicators of the presence of a clinical code, e.g., "Body dysmorphic disorder (disorder)| SCTID: 8348200". More complicated features included indicators of trends in observations or counts of features, e.g., count of psychological disorders diagnosed.

The set of features was inspired by a review of the literature and interviews with general practitioners and patients about complex mental health difficulties. The interviews were conducted as part of the wider project's qualitative work

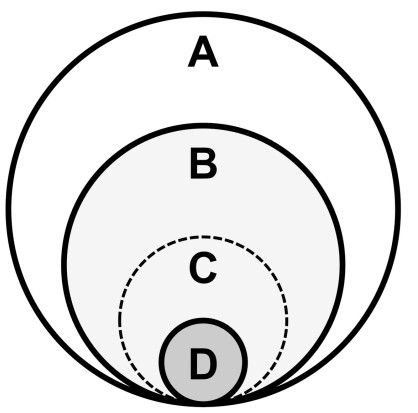

**A:** all electronic healthcare records that meet inclusion criteria.

**B:** records that meet criteria for mental ill-health.

**C:** records of people with complex mental health difficulties.

**D:** records that meet criteria for complex mental health difficulty.

**Fig 1. Visualisation of groups defined by information within electronic healthcare records.**

package, as yet unpublished at the time of writing. For this sister study, we thematically analysed the interview transcripts and produced a set of themes that included concepts about what defines complex mental health difficulties and what might indicate them. We consider these definitional and indicative concepts when developing candidate feature sets that could be operationalised and queried within a primary-care electronic health record system. The source of each feature is summarised in Table S3 where 'fs_literature' indicates a feature inspired by our literature review, 'fs_interviews' indicates a feature inspired by our interview study, and 'fs_clinician' indicates a feature inspired by the clinical members of the research team.

We also defined the following feature families that, together, accounted for all component features:

1. "Antecedent": a feature set representing features that generally precede the emergence of complex mental health difficulties in adults (e.g., child abuse, abandonment, etc), and administrative or clinical events recorded before the age of 30.

2. "Concurrent": a feature set representing concerning findings or behaviours after 30 years of age, e.g., self-harm, risk, substance misuse or dependency.

3. "Service use": a feature set representing patterns of recent use of healthcare services that indicate both intensity and variance of use, e.g., number of mental health-related SNOMED-CT codes in the patient's record.

4. "Treatment": a feature set representing patterns of therapy and prescriptions, e.g., repeated referrals to Improving Access to Psychological Therapy (IAPT).

5. "Inconsistency": a feature set representing unstable or atypical attendance activity, e.g., median count of appointments not attended, or sample entropy of appointments.

6. "Patterns of prescription": a feature set representing patterns in the prescriptions for medications of interest, e.g., the count of aborted antidepressant-medication regimes.

7. "Relevant prescriptions": a feature set indicating the presence or absence of prescriptions for our medications of interest (see Table 1).

8. "Antipsychotic prescription": a feature set containing only one feature that indicated the presence or absence of a prescription for antipsychotic medications. This recognises that much prescribing of antipsychotic medications is not for psychotic illness, but rather is for conditions such as personality disorder [17].

Each feature family was represented with five levels that indicated the count of component feature sets in a patient's record. The five levels were: 'None', indicating that none of the component features were present; 'Not none', indicating that at least the lowest observable count of the component features was present; 'Few', indicating that a lower quantile of the component features were present; 'Some', indicating that a middle quantile of the component features were present; 'Many', indicating that a higher quantile of the component features were present. Family-specific quantiles were subjectively defined based on a compromise of the research team's clinical judgement and properties of the distribution of counts of component feature sets in patients' records (See UNSEEN_create_feature_sets_appendix3.ipynb in the GitHub repository).

Finally, we defined feature-family combinations to represent all possible combinations of feature families and their levels, e.g., a feature family combination representing records with no features from the Antecedent, Concurrent, Service Use or Treatment families, but with 'Few' features from the remaining families. An additional level was permitted for each family to represent the idea that the level was irrelevant (See UNSEEN_create_feature_sets_appendix4.ipynb in the GitHub repository). For example, "A1_C0" was given a value of TRUE when a record had features from the Antecedent family but not the Concurrent family, but "A1_Cx" was given a value of TRUE when a record had features from the Antecedent family

regardless of how many Concurrent features were present. The motivation for these feature-family combinations was to represent multi-faceted, high-level, heuristic definitions of patient groups.

In summary, we looked at three levels of features - component features, feature families, and feature-family combinations - collectively called 'feature sets'. The final number of feature sets was 510,073 (Table S3).

### Ranking feature sets

All features, family feature sets, and family-combination feature sets were ranked by their two-way mutual information with the caseness variable, and scaled to the entropy of the caseness variable [28]. This provided a measure of distinguishability between cases and controls, rather than a measure that predicted either cases or controls. Our mutual-information statistic quantified the reduction in the uncertainty of caseness afforded by knowing the value of the feature set [33]. For feature sets that were continuous-valued rather than binary – e.g., sample entropy of appointments – we used the arithmetic mean of 20 runs of Ross *et al.*'s method to calculate mutual information [34]. We ranked feature sets by their scaled mutual information because there is no generalisable, analytical threshold to indicate high / good or low / poor mutual information. This means feature sets could only be interpreted as relatively better or worse, rather than absolutely good or bad.

### Evaluating and reporting

Table 2 presents the evaluation statistics. For binary feature sets, we calculated counts for all cells of the contingency tables summarising the coincidences of feature sets and the caseness variable, i.e., true positives, false positives, false negatives, and true negatives. Class-balance accuracy performs better than overall accuracy when the target variable's values are imbalanced and are weakly separable by the predictor values (the latter of which can be considered to be a measure of concept complexity and thus apt for the caseness of complex mental health difficulties) [35].

### Results

As of 22nd March 2024, the redacted and rounded count of patients in the primary-care data table of Connected Bradford was 1,155,340. The number of patients with a recent mental health disorder excluding those with severe mental illness – i.e., our study population – was 155,470 (13.5% of total population) of whom 3,040 (2.6% of the study population; 0.3% of the total population) met our criteria for caseness (Table 3).

The entropy of the caseness variable (to which feature sets' mutual information scores were scaled) was 0.099 nats, which is 14.3% of the theoretically-maximum entropy of $\approx 0.69$ nats. We also note that a randomly selected patient record from the study population would be 48-times less likely to meet our definition of complex mental health difficulties than to meet it.

Figs 2 and 3 show the distribution of values for selected feature sets. Fig 2A shows the distribution of a binary indicator of whether the set of features defining 'access to healthcare' were satisfied by the patient's record after the patient was 30 years of age. This considered repeated use of Improving Access to Psychological Therapy (IAPT), substance misuse, and relevant prescriptions. Fig 2B shows the distribution of the average annual entropy of non-attendance throughout a patient's record. Larger values indicate greater uncertainty / surprising-ness in the annual pattern of non-attendance. Fig 3A shows the distribution of 'Concurrent' family features, which were those representing concerning behaviours after 30 years of age, e.g., self-harm, substance misuse or dependency. Fig 3B shows the distribution of the 'Inconsistency' family features, which were those representing unstable or atypical attendance activity, e.g., median count of appointments not attended, or sample entropy of appointments. For all family features, cases were skewed toward larger counts of component features. A qualitative review of the distributions suggested differences between cases and control.

Despite the qualitative differences in distributions demonstrated in Figs 2 and 3, a large proportion of feature sets were non-informative, i.e., they manifested as only a single value for all patient records. Most remaining feature sets scored very low for scaled mutual information, which means they did little to improve certainty about whether a patient record met

**Table 2. Evaluation statistics and their descriptions.**

| Statistic | Description | Scope |
|---|---|---|
| Prevalence per 1,000 | The proportion of patients' records meeting the definition of the feature: $$\frac{TP+FP}{TP+FP+TN+FN} \times \frac{1}{1000}$$ | Binary feature sets |
| Mean | The arithmetic mean value of the feature. | Continuous feature sets |
| Mode | The most-frequently occurring value of the feature. | Count feature sets |
| Class Balance Accuracy | The lower bound of the average sensitivity and average positive predictive value (a.k.a. precision): $$\frac{1}{2}\left( \frac{TP}{\min\{(TP+FN),\ (TP+FP)\}} + \frac{TN}{\min\{(TN+FP),\ (TN+FN)\}} \right)$$ | Binary feature sets |
| Odds ratio | The ratio of the odds of CMHD given the presence of the feature, to the odds of CMHD given the absence of the feature. It can also be thought of as the multiplicative difference between correct and incorrect classification: $$\frac{true\ positive \times true\ negative}{false\ positive \times false\ negative}$$ | Binary, count, and continuous feature sets |
| Positive predictive value | The proportion of patients' records meeting the definition of the feature, out of all records that meet the definition of caseness: $$\frac{true\ positive}{true\ positive + false\ positive}$$ | Binary feature sets |
| Negative predictive value | The proportion of patients' records not meeting the definition of the feature, out of all records that did not meet the definition of caseness: $$\frac{true\ negative}{true\ negative + false\ negative}$$ | Binary feature sets |

a CMHD = Complex Mental Health Difficulties

b TP = True Positives

c TN = True Negatives

d FP = False Positives

e FN = False Negatives

**Table 3. Percentage of study-population records that contained the component diagnosis and medication criteria that defined caseness for complex mental health difficulties.**

| | Diagnostic criteria | % |
|---|---|---|
| Inclusion | Personality disorder OR Borderline personality disorder | 2.32 |
| | Chronic post-traumatic stress disorder OR Complex posttraumatic stress disorder | 0.08 |
| | Chronic depression OR Dysthymia OR Persistent Depressive Disorder | 1.05 |
| | Any inclusion diagnosis | 2.61 |
| Exclusion | Any exclusion diagnoses | 3.70 |
| | **Medication criteria** | **%** |
| Inclusion | Antidepressants | 60.35 |
| | Hypnotics and anxiolytics | 21.65 |
| | Psychosis-related | 5.42 |
| | Any inclusion medication | 63.52 |
| | **Both Diagnostic AND Medication criteria** | **%** |
| | Caseness | 1.96 |

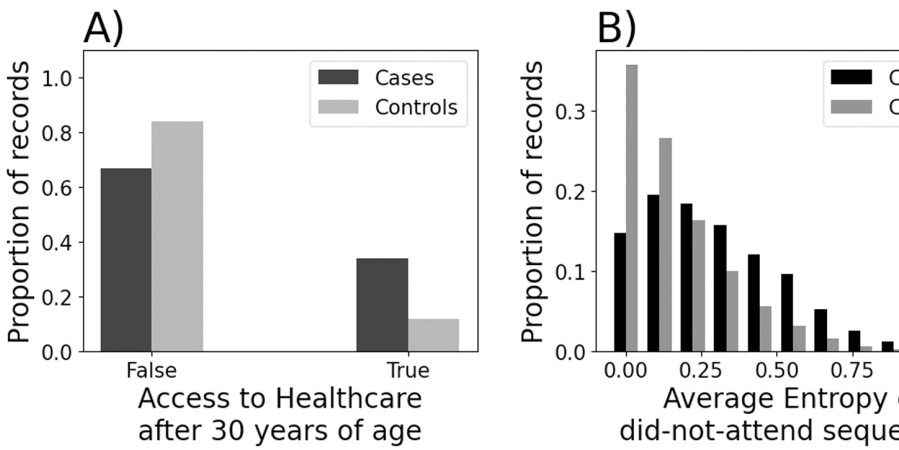

**Fig 2. Examples of the distributions of values for some component features.**

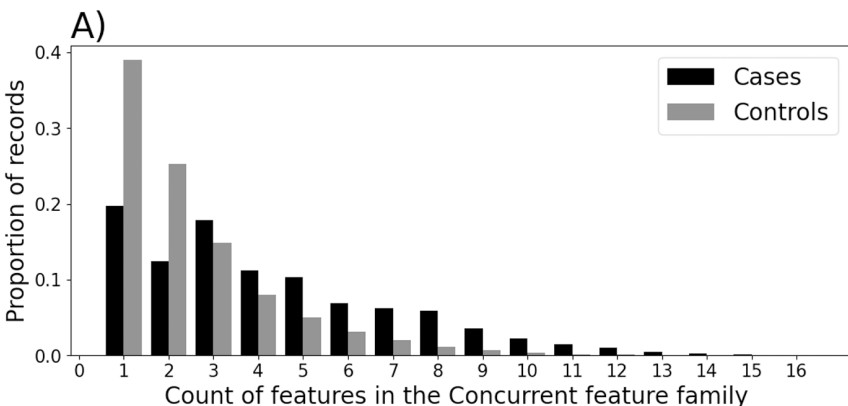

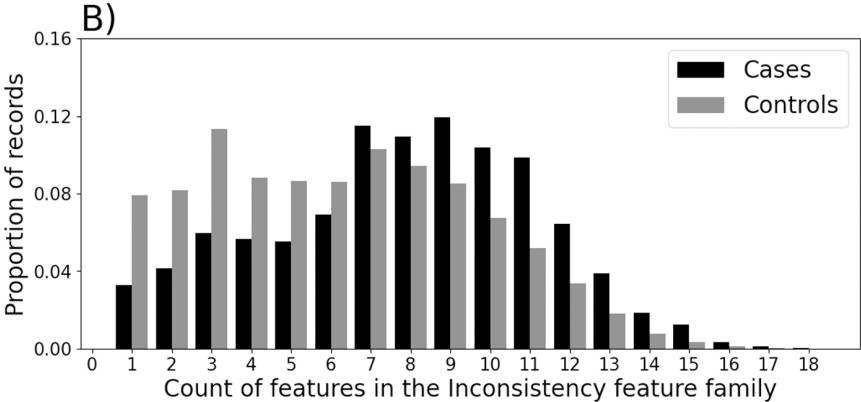

**Fig 3. Examples of the distributions of values for some feature families.**

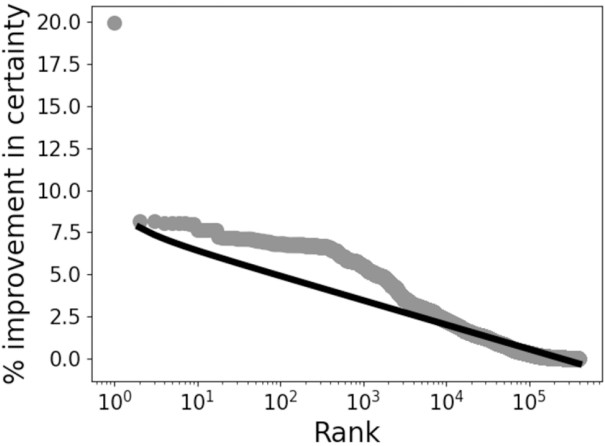

**Fig 4. Scaled mutual information for all informative feature sets in rank order.** Rank is presented in $\log_{10}$ to illustrate how tightly packed the scaled mutual information scores were across orders of magnitude of rank (illustrated by the straight-line fit). A single, outstanding feature showed a scaled mutual information value greater than 8.2%: the count of psychological disorders.

our definition of caseness. Fig 4 shows the scaled mutual information for all informative feature sets in rank order. The rank order is presented on a $\log_{10}$ scale to illustrate how tightly packed the scaled mutual information scores were across orders of magnitude of rank. Approximately $10^{5.5} = 350,000$ feature sets showed a scaled mutual information <1%, and only one feature set showed a mutual information >8.2%.

Fig 5 shows the log-scaled mutual information-by-rank plot but distinguishes component features, feature families, and feature family-combinations. All types of feature set are distributed across ranks, and all types of feature set appear more often at lower values of scaled mutual information.

The best-performing feature set was the count of psychological disorders, which showed a scaled mutual information of 19.9% — the only feature set with a scaled mutual information >8.2%. Table 4 shows the five highest-scoring feature sets, for each feature-set type.

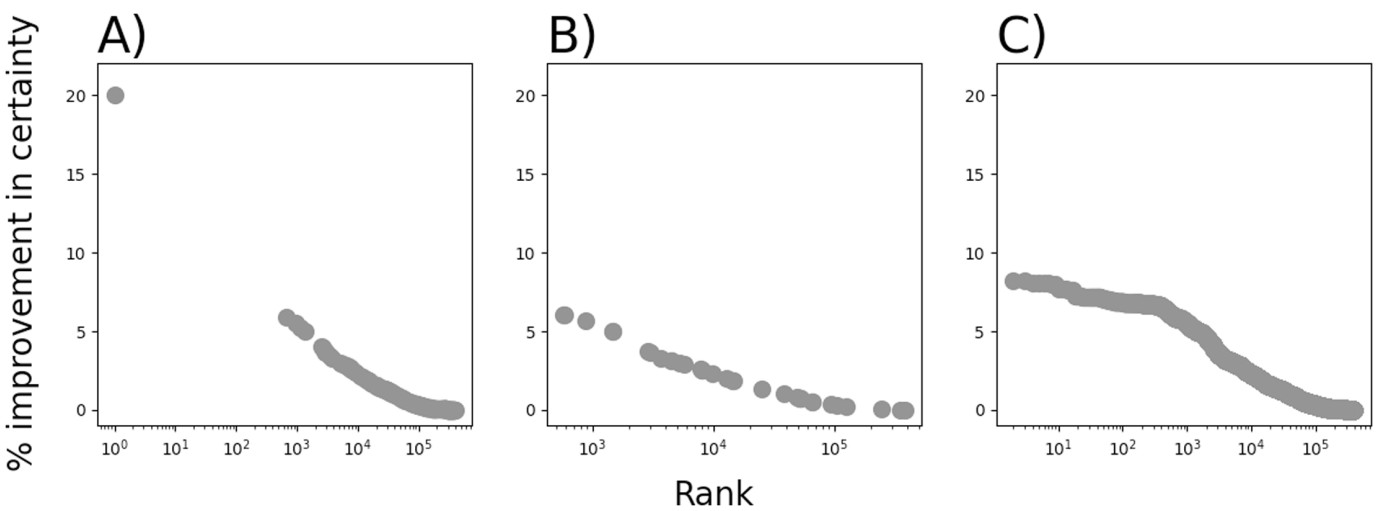

**Fig 5. Scaled mutual information for all informative feature sets in rank order. Rank is presented in $\log_{10}$.** A) Rank of component feature sets. B) Rank of family feature sets. C) Rank of feature-family combinations.

**Table 4. Evaluation statistics for the five feature sets with the highest scaled mutual information, for each feature-set type.**

| Feature set name(s) | MI$_{scaled}$ (%)[e] | Prevalence (per 1,000) | Mean | Mode | CBA[f] | OR[g] | PPV[h] | NPV[i] |
|---|---|---|---|---|---|---|---|---|
| **Component features** | | | | | | | | |
| Count of psychological disorders | 19.9 | – | – | 1 | – | 2.21 | – | – |
| Active information of non-attendance[a] | 5.8 | – | 0.22 | – | – | 3.43 | – | – |
| Count of antidepressant prescriptions | 5.5 | – | – | 0 | – | 1.05 | – | – |
| Spectral entropy of non-attendance[b] | 5.2 | – | 0.89 | – | – | <0.01 | – | – |
| Average entropy of non-attendance[c] | 4.9 | – | 0.21 | – | – | 24.53 | – | – |
| **Feature families[d]** | | | | | | | | |
| Presence of Antipsychotic Prescription | 6.0 | 18.10 | – | – | 0.58 | 15.24 | 0.21 | 0.98 |
| 1-3 Concurrent-family features | 5.6 | 856.07 | – | – | 0.08 | 0.20 | 0.01 | 0.94 |
| Presence of Treatment-family features | 4.9 | 352.83 | – | – | 0.35 | 4.25 | 0.04 | 0.99 |
| Greater than 3 Treatment-family features | 3.7 | 659.53 | – | – | 0.17 | 0.29 | 0.01 | 0.96 |
| One Treatment-family feature | 3.6 | 781.91 | – | – | 0.11 | 0.29 | 0.01 | 0.95 |
| | | | | | | | | |
| **Feature family combinations** | | | | | | | | |
| Absence of Antipsychotic Prescription family feature, with 1–3 Concurrent-family features, with or without the presence of Service Use-family features | 8.2 | 846.24 | – | – | 0.08 | 0.15 | 0.01 | 0.93 |
| Absence of Antipsychotic Prescription family feature, with 1–3 Concurrent-family features and 1 Patterns of Prescription-family feature, with or without the presence of Service Use-family features. | 8.0 | 825.8 | – | – | 0.09 | 0.15 | 0.01 | 0.93 |
| Absence of Antipsychotic Prescription family feature, with 1–3 Concurrent-family features and 1 Relevant Prescription-family feature, with or without the presence of Service Use-family features. | 8.0 | 842.5 | – | – | 0.08 | 0.15 | 0.01 | 0.93 |
| Absence of Antipsychotic Prescription family feature, with 1–3 Concurrent-family features, 1 Relevant Prescription-family feature and 1 Patterns of Prescription-family feature, with or without the presence of Service Use-family features. | 7.9 | 822.73 | – | – | 0.09 | 0.16 | 0.01 | 0.94 |
| Antipsychotic Prescription family feature, with 1–2 Antecedent-family features, 1–3 Concurrent-family features and 1 Patterns of Prescription-family feature, with or without the presence of Service Use-family features. | 7.7 | 688.29 | – | – | 0.16 | 0.16 | < 0.01 | 0.95 |

[a]Active information measured how consistent the pattern of the latest three-months of non-attendance was with the pattern of the preceding year of non-attendance.

[b]Spectral entropy quantified the uncertainty/ inconsistency of the frequency of non-attendance.

[c]Average entropy quantified the typical uncertainty/ inconsistency of quarterly non-attendance patterns.

[d]Definitions of feature families are provided in the Methods section.

[e]MI$_{scaled}$ = scaled mutual information.

[f]CBA = class balance accuracy.

[g]OR = odds ratio.

[h]PPV = positive predictive value.

[i]NPV = negative predictive value.

## Discussion

There is increasing recognition of both the mental and concomitant physical burden of mental ill-health, and  guidance for integrated care, in the UK, has been published recently [36]. Primary care clinicians will not be able to act upon this guidance for "unseen" patients that lack a recorded diagnosis of their complex needs. The work described in this article sought

to identify features within patients' electronic healthcare records that indicate complex mental health difficulties. Our aim was to generate features that might help identify patients so that they can be offered appropriate and timely care.

## The novelty of an information-theoretic approach

Statistical methods based on information theory are rare in health services research [27] (see [37] for suggested applications). We used an information-based statistic rather than a regression or classification statistic because it made fewer assumptions about the form of the relationship between feature sets and the caseness variable, e.g., directionality, linearity, distribution of residuals, relative weighting of true positives and true negatives, etc. Our mutual-information statistic quantified the reduction in the uncertainty of caseness afforded by knowing the value of the feature set.

Theoretically, mutual information is maximised for a given caseness prevalence when the value of the feature set perfectly coincides with the value of the caseness variable [33]. Information theory is suitable for medical decision making because the notion of coincidence lends itself to the binary nature of diagnoses and recording (or not) of signs and symptoms. Variance-based approaches like regression can handle this binary nature – with logistic link functions being perhaps the most-familiar trick – but, when the prevalence of the target variable is very low (as is caseness in our study), the variance of the target variable will be low, consequently. Regression analysis would have been limited in its ability to quantify the associated between the caseness of complex mental health difficulties and feature sets. This is because regression analysis would have struggled to partition / "explain" the (very little) variance of the caseness variable using the variance of the feature set. In contrast, mutual information intuitively estimated the associations by summarising the counts of patient records that showed a) evidence of both the feature set and caseness, b) no evidence of either, and c) both scenarios showing evidence of one but not the other.

## Prevalence of complex mental health difficulties

We suggest our observed prevalence of 0.3% in the population (equivalent to 2.16% of our sample of records indicating mental ill-health) represents approximately a ten-fold under-diagnosis (or under-recording of diagnoses). By comparison, Huang *et al.* suggested a population prevalence of 6.1% for all types of personality disorders, based on a global survey using face-to-face interviews of the general population [11], and Coid *et al.*'s study in Great Britain suggested a prevalence of 4.4% (weighted by the estimated prevalence of psychiatric morbidity, in 2000) [12]. Williamson *et al.* similarly found substantial under-coding of adverse childhood events in primary-care records, again, by an order of magnitude [38].

We propose three likely explanations for our low observed prevalence. The first is missingness. Routinely-collected electronic healthcare data is notorious for its poor data quality, despite frequently calls for assessment and improvement [39,40]. Second, our record-based definition requires clinicians to have clinically coded diagnoses that clinicians sometimes find difficult to diagnose and are reluctant to diagnose [41–43]. For example, chronic depression might go unrecorded explicitly, though a clinician has noted and is treating a patient's ongoing depression. In our study, a record of diagnoses of interest was the limiting criterion, with only 3% of our study population meeting the diagnosis criterion (See UNSEEN_caseness_cohort_breakdown.ipynb in the GitHub repository).

Thirdly, our records-based definition will have missed patients because it does not perfectly capture the essence of complex mental health difficulties. This is because the concept of patient complexity is difficult to precisely pin down [44–46]. In juxtaposition, the components used to define our caseness and feature definitions were coded in the prevailing nosological framework of the Diagnostic and Statistical Manual of Mental Disorders [47] and used the clinical-coding nomenclature of SNOMED-CT [48], which cater for complicated but not complex definitions. Accordingly, our definition was an example of a reductionist "diagnostic literalism" [49], which hinders the holistic approach to person-centred care that we think is needed to address complex mental health difficulties. Additionally, the healthcare-record management systems on which healthcare practitioners rely benefit from complicated yet reductionist conceptualisations because they suit

the structured / tabular data format that enables efficient storage, access, and calculation. Thus, the prevailing nosological and technical situation might have made it difficult to perfectly capture the essence of complex mental health difficulties.

Consequently, our estimates should be interpreted as a lower bound of the prevalence within the dataset, limited by the resources and processes available from the existing paradigm. Such under-coding means automated searches of databases will struggle to identifying all patients with complex mental health difficulties. This is why we sought to find informative features of complex mental health difficulties that are well-coded in routinely-collected electronic healthcare data.

### Features informativeness

We suggest that the features that were non-informative (i.e., manifesting only a single value in all records) is explained by the fact that most features were highly-specified binary variables. Despite generally poor informativeness of features, the count of psychological disorders recorded in a patient's record had a scaled mutual information of 19.9%. It is perhaps not surprising that information is shared between the count of psychological disorders and our definition of the caseness of complex mental health difficulties because records met our definition of caseness by 1) including a diagnosis of at least one of the six high-level disorder groups and their taxonomic children, and by 2) including a prescription for at least one of the 16 medications used to treat a variety of mental ill-health disorders. Both this feature and our definition of caseness might be representations of the same underlying concept of comorbid mental disorders.

We noted that the probability and the odds of a patient record meeting our definition of complex mental health difficulties monotonically increases with every additional diagnosis recorded. Thus, the count of psychological disorders appears to be a "dose"-dependent proxy for the likelihood of meeting our definition of complex mental health difficulties. This "dose"-dependent proxy echoes tools like Charlson Comorbidity Index, INTERMED and LOCUS, which are used to stratify patients by their "complexity", as a proxy for the level of care they are expected to require [46]. Like the count of psychological disorders, the Charlson Comorbidity Index is sum of conditions, but each condition is weighted [50]. INTERMED assesses patients' biopsychosocial complexity [51,52], while LOCUS assesses psychiatric and chemical-dependency problems with a focus on defining the level of care needed [53]. Both INTERMED, LOCUS and others require surveys, interviews, or self-assessment but our count of psychological disorders is a simple rule easily implemented in patient management software. The count of psychological disorders might be a useful dose-dependent indicator of complex mental health difficulties that can be calculated easily and automatically in electronic health record systems.

Further support for the validity of this feature come from Newman *et al.* who note that comorbid mental disorders accompany "*complications that challenge treatment planning, compliance, and coordination of service delivery*" [54], which aligns with descriptions of complex mental health difficulties [2,3]. The idea of a general psychopathology factor is founded on psychological comorbidity, also [55]. Similar to our comments on the (in)appropriateness of the current nosological paradigm, many have argued to move beyond a simple, cumulative / additive model of patient complexity (e.g., [56]) and suggest non-linear, network-based conceptions of pathology, nosology, and emergent burden [49,57,58]. Therefore, we encourage further study of how patterns of psychological disorder might be informative of complex mental health difficulties.

### Other component features, feature families, and feature-family combinations

Apart from the outstanding performance by the count of psychological disorders, three of the top-five component features were entropy measures of non-attendance patterns, from the Inconsistency family of features. One should keep in mind that chronic non-attendance would be a consistent (albeit concerning) behaviour. If we consider these Inconsistency-family features together, we might summarise that caseness was indicated by persistent inconsistency in quarterly patterns of non-attendance, but not in the frequency spectrum. Non-attendance has been associated with complex psychosocial difficulties [59], poorer social functioning [60], and lower socioeconomic status [61,62], which might reasonably contribute to the complexity of patient's mental health difficulties. But, it must be noted that the scaled mutual information was low, at only 4.9–5.8%.

Considering feature families, the top rank of the Antipsychotic Prescription family is noteworthy by the higher-than-average positive predictive value associated with its presence in a patient's record (despite its low class balance accuracy). Antipsychotics are mainly prescribed for schizophrenia and bipolar disorder, but we excluded all records that contained these diagnoses from our analysis. Excess antipsychotics prescriptions due to uncoded psychosis is unlikely because registries of severe mental illness are expected to be maintained by primary-care practice in the UK [63] and recorded rates in the UK are consistent with epidemiological studies [64]. Similarly, excess prescriptions for the other diagnoses that constituted our definition is unlikely because our observed prevalence of antipsychotic prescriptions was over twice the prevalence of having any diagnosis of interest (5.42 versus 26.1 per 1,000 records). Given that antipsychotics are routinely used for mood stabilisation of patients with comorbid personality disorder [65–67], we suggest that the discrepancy between our observed diagnoses and antipsychotic prescription might indicate treatment for under-coded diagnoses indicative of complex mental health difficulties.

Feature family combinations scored better than component features and feature families, though scores were low. The common sub-combination (and highest scorer) was no antipsychotic prescriptions, with 1–3 Concurrent-family features, with or without the presence of Service Use-family features. Records with this sub-combination were less likely to meet our definition of caseness. This implies that, although patients are currently experiencing some of our candidate features for complex mental health difficulties, there are not likely to meet our definition if they are not taking antipsychotic medicines.

## Limitations

We did not adjust for confounding or collider bias when calculating mutual information [68,69]. To do so would have required us to encode all features that are part of the system under study and have hypothesised all informational relationships between all features. Only then could the problems of confounding and collider bias have been judiciously addressed by conditioning [69]. Instead, we opted to minimise assumptions about the relationship between variables, which is why, for example, we describe our odds ratios as being in the context of all unmeasured confounding and bias. Thus, some feature sets might score artificially low by way of the reversal paradox [70].

It must also be noted that Ross *et al.*'s method to calculate mutual information relies on a nearest-neighbour method that involves a randomisation step [34]. Therefore, the mutual information for component features with count and continuous values varied with every run. As noted earlier, we choose to take the arithmetic mean of 20 runs of Ross *et al.*'s method. We investigated the spread of calculated values from 200 runs for an arbitrarily selected feature and found the standard deviation of scaled mutual information was 2.49% and the interquartile range was 3.56% (see UNSEEN_create_feature_sets_base.ipynb in the GitHub repository). This spread of possible values could result in substantial reordering of the ranks of count- and continuous-valued feature components, but never enough to have competed with the only outstanding feature representing the count of psychological disorders on record.

Finally, our codelists contained 102 of the "sensitive" SNOMED-CT codes that were not included in the Connected Bradford primary-care database. These missing codes referred to specific forms of abuse and might have resulted in fewer records being identified for related feature sets in the Antecedent and Concurrent feature families. We cannot know the extent to which this affected the informativeness of relevant feature sets but our exhaustive codelists are likely to have identified related codes within patients' records.

## Contribution to knowledge

Little is known about the recognition of complex mental health difficulties in primary care. Our study leveraged the existing infrastructure of electronic healthcare records and clinical coding, founded on insights from the experiences of patients and healthcare professionals gathered from its sister qualitative study. We did this with the novel application of an information-theoretic evaluation of multi-level, multi-dimensional feature sets.

The prevalence of complex mental health difficulties as per our definition was low, which made it difficult for any feature to be informative. Almost none of the features derived from information within electronic healthcare records were notably informative of our definition. These findings support the idea that complex mental health difficulties are difficult to identify and operationalise, leading to an under-recording of cases that limits the use of electronic healthcare records to support identification, study, and provision of appropriate and timely care.

The count of psychological disorders was a lone, outstanding feature with a definition similar to our definition of complex mental health difficulties, and with reasonable theoretical association. Other component features, feature families, and feature-family combinations variously but marginally indicated in favour and against caseness. We interpret these findings under the assumption that diagnoses indicating complex mental health difficulties are vastly under-coded. To have identified indicators of complex mental health difficulties in the presence of such under-coding makes us optimistic that more indicators with greater strength could be identified with better recognition and recording. This is why our mixed-methods research study entitled "Understanding Services for people with Complex Mental Health Difficulties (UnSeen)" developed recommendations and a toolkit to help practices and new services work together. We hope our findings motivate improvements in diagnostic frameworks and patient-record management systems that better handle the unseen reality of complex mental health difficulties.

## Supporting information

**S1 Table. List of "sensitive" clinical codes that were excluded from the Connected Bradford primary-care database at the time of this study.**
(CSV)

**S2 Table. URLs for opencodelist.org codelists developed and used by the UnSeen project team.**
(CSV)

**S3 Table. Full list of features and their definitions.**
(CSV)

**S4 Table. Basic demographic statistics of cases and controls.** Readers are advised not to over-interpret these kinds of tables to conclude that the cases and controls from our study are representative of cases and controls, generally; that any overlap in distributions has implications for "significant" differences; and that any difference observed in our sample indicate distinguishing features. These errors are described as the Table 1 fallacy (https://doi.org/10.2106/JBJS.21.01166) and the Table 2 fallacy (https://doi.org/10.1093/aje/kws412) (see also "Out of balance" by Darren Dahly for a less formal discussion of the Table 1 fallacy; doi: https://statsepi.substack.com/p/out-of-balance).
(CSV)

## Acknowledgments

This study is based on data from Connected Bradford (REC 18/YH/0200 & 22/EM/0127). The data is provided by the citizens of Bradford and district, and collected by the National Health Service (NHS), UK Department of Education (DfE) and other organisations as part of their care and support. The interpretation and conclusions contained in this study are those of the authors alone. The NHS, DfE and other organisations do not accept responsibility for inferences and conclusions derived from their data by third parties.

## Author contributions

**Conceptualization:** Phillip Oliver, Michelle Horspool, Vyv Huddy, Christopher Burton.

**Data curation:** Ciarán McInerney.

**Formal analysis:** Ciarán McInerney.

**Funding acquisition:** Phillip Oliver, Michelle Horspool, Vyv Huddy, Christopher Burton.

**Investigation:** Ciarán McInerney.

**Methodology:** Ciarán McInerney, Christopher Burton.

**Project administration:** Ciarán McInerney.

**Supervision:** Christopher Burton.

**Visualization:** Ciarán McInerney.

**Writing – original draft:** Ciarán McInerney.

**Writing – review & editing:** Ciarán McInerney, Phillip Oliver, Ada Achinanya, Michelle Horspool, Vyv Huddy, Christopher Burton.

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
