## [Decision Letter · Decision Letter 0]

4 Mar 2024

PONE-D-24-01070Identifying primary-care features associated with complex mental health difficultiesPLOS ONE

Dear Dr. McInerney,

Thank you for submitting your manuscript to PLOS ONE. After careful consideration, we feel that it has merit but does not fully meet PLOS ONE’s publication criteria as it currently stands. Therefore, we invite you to submit a revised version of the manuscript that addresses the points raised during the review process.

 Please submit your revised manuscript by Apr 18 2024 11:59PM. If you will need more time than this to complete your revisions, please reply to this message or contact the journal office at plosone@plos.org . Please include the following items when submitting your revised manuscript:

We look forward to receiving your revised manuscript.

Kind regards,

Enzo Pasquale Scilingo, Ph.D.

Academic Editor

PLOS ONE

Journal Requirements:

"Funded by the National Institute for Health Research Research for Patient Benefit - "Mental Health in the North" call ( NIHR203473)"

**Additional Editor Comments:**

The manuscript needs to be improved before it can be recommended for publication. In particular, some issues about the protocol design need to be carefully addressed. 

Reviewers' comments:

Reviewer's Responses to Questions

**Comments to the Author**

1. Is the manuscript technically sound, and do the data support the conclusions?

Reviewer #1: Partly

Reviewer #2: Yes

2. Has the statistical analysis been performed appropriately and rigorously? 

Reviewer #1: Yes

Reviewer #2: I Don't Know

3. Have the authors made all data underlying the findings in their manuscript fully available?

Reviewer #1: No

Reviewer #2: Yes

4. Is the manuscript presented in an intelligible fashion and written in standard English?

Reviewer #1: Yes

Reviewer #2: Yes

5. Review Comments to the Author

Reviewer #1: This is an interesting attempt to identify cases of complex mental illness in medical records. While not completely successful in doing so, it does add to the literature in this area. However, I have concerns regarding the inclusion and exclusion criteria, in particular the way that certain items were operationalised. I feel "chronic depression" will not have captured many with chronic depression; while "severe mental illness" excluded many incentivised terms for psychotic illness, and only four antipsychotics were considered. I feel these issues require some more detailed justification and explanation in the text. Specific comments in relation to these points and others are below. There were also quite a few editorial issues, such as the lack of graph titles/footnotes, mislabelled supplements and tables, and the broken link to github.

1. Line 63: The prevalence appears to vary substantially in the WHO study with >7% in the US And Colombia and only 2% in Western Europe. This study in the UK suggests around 4% which may be more accurate: https://pubmed.ncbi.nlm.nih.gov/16648528/

2. Lines 68-72: This study uses THIN, which does not have full UK coverage. The methods of this paper state at the time of the study it had records for 744 GP practices, accounting for approximately 6% of the UK population. Therefore, if you assume a population of 60 million, the denominator should be 6% of this, and this study identified around 1.3% of patients with codes for PD (46,210 / 3,600,000), rather than 0.008%. Likewise, the Norwegian study was only looking at diagnoses made in a one year period, and so was not looking at the prevalence of PD in the sample, but those who had a consultation for PD in that one year.

3. Line 73: This study is based in the US which is a very different EHR system to ours. Depression (as a clinical code or prescription of antidepressants) is recorded relatively well in the UK, it is the determining whether it is a persistent depression that is harder. This may be a better reference for how much is recorded for depression/PTSD in the UK, though it doesn’t detail persistent depression: https://bmcmedinformdecismak.biomedcentral.com/articles/10.1186/s12911-023-02296-z/tables/3

4. Line 81: Another really important reason the authors may want to consider is that as of Jan ’24 NHS England advise “ICSs are strongly encouraged to consider the physical health needs of all people severely affected by their mental illness…This includes, but is not limited to, those with a diagnosis of personality disorder, eating disorder or severe depression, and people with mental health rehabilitation needs”. However, without these diagnoses recorded in primary care the GP will be unable to act upon this: https://www.england.nhs.uk/long-read/improving-the-physical-health-of-people-living-with-severe-mental-illness/#the-smi-register-and-smi-annual-physical-health-checks

5. Line 110: Note, this link does not currently work so I have been unable to review anything on the github.

6. Line 161: This data is in S2 Table, not S1.

7. Table 1: This data is in S2 Table, not S3.

8. Table 1: dementia is listed as an exclusion but not justified in the text.

9. Line 161 you state “a SNOMED-CT diagnostic code for caseness (see link associated with "Mental disorder | SCTID: 74732009 + child codes" but then in Table 1 you have much more stringent inclusion criteria. Referring to Figure 1 I see that this is a stepped approach, and assume a case is a patient with a diagnosis as per the inclusion in Table 1? This could be clearer in the text.

10. Controls: Were controls those that met neither the diagnosis or the medication criteria, or could they meet one criteria but not the other? Were controls able to have diagnoses such psychotic disorders? We know a lot of patients with codes for psychosis go on to receive a code of schizophrenia, and so I wonder whether removing them may have given a “cleaner” control group. GPs in England are incentivised to keep a record of patients with SMI (including schizophrenia and bipolar disorder but also other psychoses – e.g. schizoaffective schizophrenia, first episode psychosis). Because of this, people with any of the SNOMED codes included in the NHS incentivised definition are likely to be well recorded, but not all meet your criteria of SMI. The Hardoon paper from your introduction found that of people with personality disorder, 20% had an SMI (schizophrenia, bipolar disorder, other psychosis) diagnosis, so the cross-over between SMI and PD needs some careful thought. Excluding them from cases removes potentially quite a few people with PD, while not fully excluding them from controls (i.e. including those with other psychoses) may be contaminating your control group.

11. Table 1: Could the authors justify their selection of antipsychotics? I was interested that only four were chosen. While quetiapine, olanzapine and risperidone are very commonly prescribed, the paper by Hardoon et al lists chlorpromazine, aripirazole and haloperidol as more commonly prescribed to patients with personality disorder than flupentixol, while more recent analysis of patients with SMI shows that aripiprazole is now the second most commonly prescribed.

12. Line 190: Could the authors justify the age 30 as a cut off for “antecedent” events? If a patient has an “antecedent” event such as ADHD under 30, do they automatically also have a concurrent event for ADHD? As obviously diagnoses such as these may be recorded by the GP very infrequently, but still be present.

13. Line 269: Could the authors also report the exact number of controls? Some basic descriptive information on cases and controls would be useful here.

14. Line 276: It is not clear which feature sets are shown in these figures.

15. Figure 2a: What does the true/false relate to?

16. Figure 3a: Is this a count of all figures?

17. Figure 5: What do a, b and c refer to? I think the titles/footnotes for these figures are missing?

18. There is no table 4 in the manuscript, and S3 Table is not referenced anywhere.

19. Table 5: While SMI is defined here, I think the use of the same abbreviation to mean two different things (severe mental illness and scaled mutual information) is going to confuse readers.

20. Table 5: I was surprised you excluded obesity, and there was no consideration of smoking. I would have thought, alongside alcohol and drug misuse these would be useful predictors of mental health conditions. While I agree these are not specific to complex mental illness, possibly in combination with other things they may have been useful.

21. Line 329: While I agree there is a great deal of under-reporting/under-diagnosis in primary care; you did exclude patients with SMI and we know co-occurrence is common (around 20% in the Hardoon paper). The WHO prevalence calculated by Huang et al is in adults, whereas I think you are calculating yours over the whole population, and Huang et al estimated a prevalence far lower for Western Europe (2.4%). This study in the UK suggests around 4% which may be more accurate: https://pubmed.ncbi.nlm.nih.gov/16648528/. Could the authors take some of these nuances into consideration? That being said, I think the biggest under-reporting is “chronic depression”. I suspect that is because this is rarely coded for. Other colleagues have instead looked at persistence of standard depression codes over time, or long-term prescription of antidepressants. This means many with chronic depression are potentially in your control population. There are also SNOMED CT codes such as “persistent depressive disorder” which suggest chronic depression.

22. Line 424: But you didn’t include “other psychosis” in your exclusions, and therefore there could well be people with psychosis in your cases cohort. The list of SNOMED codes used for incentivisation is here, if you select PCD refsets” and set Cluster ID as “MH_COD”. You will see many conditions that you did not exclude which may warrant antipsychotic prescription: https://app.powerbi.com/view?r=eyJrIjoiNDRmYjEwMzQtZGE3MS00ZGE5LTgwMTUtNjQ2NGE1NTZiYmEzIiwidCI6IjM3YzM1NGIyLTg1YjAtNDdmNS1iMjIyLTA3YjQ4ZDc3NGVlMyJ9 .

23. General: I wonder if it warrants a comment regarding validation in other datasets?

Reviewer #2: This manuscript studies health records to assess What features within patients’ primary-care

electronic healthcare records might indicate that a patient has complex mental health difficulties .

Complex mental health difficulties is defined as difficulties more persistent or disruptive than the common mental disorders but which do not meet current definitions of severe mental illness like psychosis or bipolar disorder. It is an interesting paper that tries to understand what distinguishes a complex case from a non-complex case.

They also used a large dataset including all patients registered for at least 1 year in a practice and have some indication of mental illness because of diagnoses or medication. Severe mental illness diagnoses were excluded.

I have problems with understanding the cases in their case-control design. Cases were those records demonstrating recent, active, complex mental health difficulties, defined by having 1) a SNOMED-CT diagnostic code for caseness from at any time prior to 31st December 2021 and 2) a prescription for medicines of interest within ten years prior to 31st December 2021.

- What happens with patients that did not have registrations for that period of time ? Because they were registered in a practice for less than 10 years? Or does the dataset also includes data from there patients from their former practices?

- Why is this defined as cases? Why not also the ones with only a prescription \? As the authors mention GPs might not always be accurate in registering the diagnoses. So including this group with only prescription as control might not really be a good control.

- Cases could also be defined as the ones with having longer period of time of either registration of diagnoses or prescriptions. As the definition suggest a persistency of symptoms.

The authors use an information-theoretic approach. This could explained more in the introduction; why is that innovative? They discuss it in the discussion but this could be introduced more in the first part.

They could explain more why it is important to know the features of registered complex mental health problems. As they already mentioned that registration is not accurate. So what does it tell us these features?

The findings support the idea that complex mental health difficulties are difficult to identify and operationalise, leading to an underecording of cases that limits the use of electronic healthcare records to support identification, study, and provision of appropriate and timely care. Partly I agree that GPs under record mental health problems. But in present study about 13% had either a diagnosis or a prescription. How is that in line with prevalence in the general population?

Why for the concurrent feature findings or behaviours are concurrent after 30 years of age, e.g. self-harm, risk, substance misuse or dependency? What about patients younger than 30?

6. PLOS authors have the option to publish the peer review history of their article (what does this mean? ). If published, this will include your full peer review and any attached files.

**Do you want your identity to be public for this peer review?** For information about this choice, including consent withdrawal, please see our Privacy Policy .

Reviewer #1: No

Reviewer #2: No

---

## [Author Response · Author response to Decision Letter 1]

28 May 2024

Dear, Reviewers.

We thank you for your time and expertise that you committed to reviewing our submission. We have amended our manuscript in response to your insightful comments. We provide a response to each of your comments, below.

Yours sincerely,

Ciarán McInerney

in co-authorship with the UnSeen project team

Reviewer #1: This is an interesting attempt to identify cases of complex mental illness in medical records. While not completely successful in doing so, it does add to the literature in this area. However, I have concerns regarding the inclusion and exclusion criteria, in particular the way that certain items were operationalised. I feel "chronic depression" will not have captured many with chronic depression; while "severe mental illness" excluded many incentivised terms for psychotic illness, and only four antipsychotics were considered. I feel these issues require some more detailed justification and explanation in the text. Specific comments in relation to these points and others are below. There were also quite a few editorial issues, such as the lack of graph titles/footnotes, mislabelled supplements and tables, and the broken link to github.

1. Line 63: The prevalence appears to vary substantially in the WHO study with >7% in the US And Colombia and only 2% in Western Europe. This study in the UK suggests around 4% which may be more accurate: https://pubmed.ncbi.nlm.nih.gov/16648528/

- We thank the reviewer for directing us to the additional reference. We have included it in the manuscript.

2. Lines 68-72: This study uses THIN, which does not have full UK coverage. The methods of this paper state at the time of the study it had records for 744 GP practices, accounting for approximately 6% of the UK population. Therefore, if you assume a population of 60 million, the denominator should be 6% of this, and this study identified around 1.3% of patients with codes for PD (46,210 / 3,600,000), rather than 0.008%. Likewise, the Norwegian study was only looking at diagnoses made in a one year period, and so was not looking at the prevalence of PD in the sample, but those who had a consultation for PD in that one year.

- We thank the reviewer for spotting the particular sampling bias of T.H.I.N. that is noted on page 2 of the cited UK study. For clarity, we arrived at our rounded value of 0.08% by using the 46,210 count of those recorded with personality disorder as the numerator, and 60 million as the denominator. We agree that the reviewer’s suggested denominator is more appropriate, so we have changed the estimated prevalence.

- Regarding the Norwegian study, we have amended the text to clarify that the prevalence is a one-year incidence rather than a prevalence.

3. Line 73: This study is based in the US which is a very different HER system to ours. Depression (as a clinical code or prescription of antidepressants) is recorded relatively well in the UK, it is the determining whether it is a persistent depression that is harder. This may be a better reference for how much is recorded for depression/PTSD in the UK, though it doesn’t detail persistent depression: https://bmcmedinformdecismak.biomedcentral.com/articles/10.1186/s12911-023-02296-z/tables/3

- We thank the reviewer for directing us to the alternative reference.

4. Line 81: Another really important reason the authors may want to consider is that as of Jan ’24 NHS England advise “ICSs are strongly encouraged to consider the physical health needs of all people severely affected by their mental illness…This includes, but is not limited to, those with a diagnosis of personality disorder, eating disorder or severe depression, and people with mental health rehabilitation needs”. However, without these diagnoses recorded in primary care the GP will be unable to act upon this: https://www.england.nhs.uk/long-read/improving-the-physical-health-of-people-living-with-severe-mental-illness/#the-smi-register-and-smi-annual-physical-health-checks

- We thank the reviewer for highlighting the advice given in January 2024. Our draft missed this information because the study was completed in 2023. We now refer to this important publication in our Discussion, to set the context.

5. Line 110: Note, this link does not currently work so I have been unable to review anything on the github.

- We apologise for the inconvenience. We have contacted Connected Bradford, who have now made all study files public.

6. Line 161: This data is in S2 Table, not S1.

- Apologies for this error, and thanks to the reviewer for spotting it. We have amended it.

7. Table 1: This data is in S2 Table, not S3

- Apologies for this error, and thanks to the reviewer for spotting it. We have amended it.

8. Table 1: dementia is listed as an exclusion but not justified in the text.

- Thanks for pointing out our oversight (this had been included in earlier documents). We have now added our justification to the others.

9. Line 161 you state “a SNOMED-CT diagnostic code for caseness (see link associated with "Mental disorder | SCTID: 74732009 + child codes" but then in Table 1 you have much more stringent inclusion criteria. Referring to Figure 1 I see that this is a stepped approach, and assume a case is a patient with a diagnosis as per the inclusion in Table 1? This could be clearer in the text

- Apologies for this error, and thanks to the reviewer for spotting it. We have made the following amendments: the explanation of the definition of our study population now includes the note about “Mental disorder | SCTID: 74732009 + child codes” in S2 Table; the explanation of our caseness now reads “…defined by having 1) at least one SNOMED-CT diagnostic code from our inclusion list, excluding those from our exclusion list…(Table 1)”.

10. Controls: Were controls those that met neither the diagnosis or the medication criteria, or could they meet one criteria but not the other? Were controls able to have diagnoses such psychotic disorders? We know a lot of patients with codes for psychosis go on to receive a code of schizophrenia, and so I wonder whether removing them may have given a “cleaner” control group. GPs in England are incentivised to keep a record of patients with SMI (including schizophrenia and bipolar disorder but also other psychoses – e.g. schizoaffective schizophrenia, first episode psychosis). Because of this, people with any of the SNOMED codes included in the NHS incentivised to keep a record of patients with SMI (including schizophrenia and bipolar disorder but also other psychoses – e.g. schizoaffective schizophrenia, first episode psychosis). Because of this, people with any of the SNOMED codes included in the NHS incentivised definition are likely to be well recorded, but not all meet your criteria of SMI. The Hardoon paper from your introduction found that of people with personality disorder, 20% had an SMI (schizophrenia, bipolar disorder, other psychosis) diagnosis, so the cross-over between SMI and PD needs some careful thought. Excluding them from cases removes potentially quite a few people with PD, while not fully excluding them from controls (i.e. including those with other psychoses) may be contaminating your control group

- We thank the reviewer for this comment, which we think is related to comment 22. We decided to replace our codelists for bipolar and schizophrenia with a single codelist for the NHS ResSet for MH-COD, which specifies psychosis, schizophrenia and bipolar affective disease codes (https://www.opencodelists.org/codelist/nhsd-primary-care-domain-refsets/mh_cod/20210127/#tree). The ramifications were of this change are:

• a redacted-and-rounded count of 5,500 patient records contained the MH-COD codes, whereas only 4,040 records contained either bipolar (1,860) or schizophrenia (2,180).

• The percentage of caseness among those with mental ill-health in the Connected Bradford dataset dropped to 1.96% from 2.16%.

• The percentage of patient records with at least one diagnosis dropped to 2.6% from 2.9%.

• The percentage of patient records with at least one medication dropped to 61.5% from 62.6%.

• The caseness variable’s entropy dropped to 0.099 nats from 0.106 nats.

• The caseness variable's entropy dropped to 14.3% of its theoretical maximum, from 15.4% of its theoretical maximum.

• The accuracy (a.k.a. hit rate) of a rule classifying all study-population records as meeting our definition of complex mental health difficulties dropped from 2.218% to 2.029%.

• The accuracy (a.k.a. hit rate) of a rule classifying none of study-population records as meeting our definition of complex mental health difficulties increased from 97.782% to 97.971%.

• The odds that record did not meet our definition of complex mental health difficulties increased from 44 to 48-times as likely.

- We also recognise the cross-over between SMI and PD, both in terms of shared risk factors and overlapping symptoms and impairment. Nevertheless, we exclude selective SMI codes because one of the potential predictors we are using is antipsychotic medication in the absence of psychosis. Pragmatically also, people with co-morbid PD and psychosis are likely to be visible to specialist mental health services rather than “unseen”.

11. Table 1: Could the authors justify their selection of antipsychotics? I was interested that only four were chosen. While quetiapine, olanzapine and risperidone are very commonly prescribed, the paper by Hardoon et al lists chlorpromazine, aripiprazole and haloperidol as more commonly prescribed to patients with personality disorder than flupentixol, while more recent analysis of patients with SMI shows that aripiprazole is now the second most commonly prescribed.

- We thank the reviewer for pointing out this omission. We have included the following medications in the updated scripts: chlorpromazine, aripiprazole and haloperidol. Note: we first applied the suggested change to the exclusion codelist, and have presented the ramifications previously in our response. The ramifications of adding these medications to the caseness definition are:

• a redacted-and-rounded count of patient records contained medications for psychosis and related medications increased from 7,610 to 8,430. This did not change the percentage of records meeting this component of our caseness definition, at two decimal places.

• The percentage of caseness among those with mental ill-health in the Connected Bradford dataset remained at 1.96%. This was unsurprising given that we have noted how the diagnoses criterion has been the limiting component of our definition.

• The count of records meeting our criteria for caseness remained unchanged at 3,040.

• The percentage of patient records with at least one medication remained at 61.5%, but the redacted-and-rounded count decreased from 68,210 to 68,060. This indicates that the addition of the medications to the list has updated information on polypharmacy, even though it has not identified any new records.

• The caseness variable’s entropy remained at 0.099 nats.

• The caseness variable's entropy remained at 14.3% of its theoretical maximum.

• The accuracy (a.k.a. hit rate) of a rule classifying all study-population records as meeting our definition of complex mental health difficulties increased 2.029% to 2.032%.

• The accuracy (a.k.a. hit rate) of a rule classifying none of study-population records as meeting our definition of complex mental health difficulties decreased from 97.971 to 97.968%.

• The odds that record did not meet our definition of complex mental health difficulties remained unchanged at 48-times as likely.

12. Line 190: Could the authors justify the age 30 as a cut off for “antecedent” events? If a patient has an “antecedent” event such as ADHD under 30, do they automatically also have a concurrent event for ADHD? As obviously diagnoses such as these may be recorded by the GP very infrequently, but still be present.

- We recognise the reviewer’s point about features present across the life-course. Crucially, the unit of analysis in our study is the record, not the person or healthcare phenomena. For the example given by the reviewer, the signal we are interested in is a recorded diagnosis of ADHD before the age of 30, rather than the presence of the patient’s ADHD. The patient’s ADHD will have existed undiagnosed before the diagnosis and will continue to exist afterward. Precise dates of onset (and magnitude of) mental or psychological conditions are not possible so, rather than assume diagnoses are a proxy for such things, we have tried to be clear that we are studying records rather than patients. Additionally, any age threshold is arbitrary; we used <18 as child or young person, 18-29 as emerging adult, and 30 or older as adult.

13. Line 269: Could the authors also report the exact number of controls? Some basic descriptive information on cases and controls would be useful here.

- We now provide this information in a classic Table 1, as supplementary material: S4 Table. We didn’t provide a classic table 1 showing distributional summary statistics of immutable demographics of cases and controls because such information does not help us to achieve our aim. Particularly, our study adopted a predictive rather than explanatory paradigm so concerns like confounding (which some think to be highlighted in a classic table 1) are not issues (see doi: 10.1214/10-STS330). Furthermore, readers inevitably over-interpret these kinds of tables to conclude that the cases and controls from our study are representative of cases and controls, generally; that any overlap in distributions has implications for “significant” differences; and that any difference observed in our sample indicate distinguishing features. These errors are described as the Table 1 fallacy (doi: 10.2106/JBJS.21.01166) and the Table 2 fallacy (doi: 10.1093/aje/kws412) (see also "Out of balance" by Darren Dahly for a less formal discussion of the Table 1 fallacy; doi: https://statsepi.substack.com/p/out-of-balance).

14. Line 276: It is not clear which feature sets are shown in these figures.

- Apologies for the lack of clarity. We submitted figures with accompanying metadata using the online system. It appears they were not carried through to the version sent for review. The figures were expected to be viewed with the accompanying caption. While the horizontal axis of figure 2B is labelled as average entropy, we acknowledge that the other figures are not self-explanatory. We have amended the x-axes to clearly state what the features are. For figure 2B, the updated caption is “Average entropy of did-not-attend sequences”.

15. Figure 2a: What does the true/false relate to?

- As mentioned in point 14, we apologise for the inconvenience and we have amended the x-axes to clearly state that the feature is a binary indicator indicating whether the set of features defining ‘access to healthcare’ were satisfied by the patient’s record after the patient was 30 years of age.

16. Figure 3a: Is this a count of all figures?

- As mentioned in point 14, we apologies for the inconvenience and we have amended the x-axes to clearly state that the feature is the count of ‘Concurrent’ family features, which are intended to represent concerning behaviour after 30 years of age, e.g. self-harm, substance misuse or dependency.

17. Figure 5: What do a, b and c refer to? I think the titles/footnotes for these figures are missing?

- The reviewer is correct to note that the captions for all figures seem to have been missing from the document they received. We apologise for our part in the inconvenience. Figure 5 shows the scaled mutual information for all informative feature sets in rank order. Rank is presented in log10. A = component features. B = feature families. C = feature family combinations.

18. There is no table 4 in the manuscript, and S3 Table is not referenced anywhere.

- Apologies for this error, and thanks to the reviewer for spotting it. All references to Table 5 have been replaced with reference to Table 4. Reference to Table S3 has now been included in the Methods subsection “Features and feature sets” where originally intended.

19. Table 5: While SMI is defined here, I think the use of the same

---

## [Decision Letter · Decision Letter 1]

25 Feb 2025

PONE-D-24-01070R1Identifying primary-care features associated with complex mental health difficultiesPLOS ONE

Dear Dr. McInerney,

Thank you for submitting your manuscript to PLOS ONE. After careful consideration, we feel that it has merit but does not fully meet PLOS ONE’s publication criteria as it currently stands. Therefore, we invite you to submit a revised version of the manuscript that addresses the points raised during the review process. The manuscript needs minor revisions before it can be recommended for publication. Please submit your revised manuscript by Apr 11 2025 11:59PM. If you will need more time than this to complete your revisions, please reply to this message or contact the journal office at plosone@plos.org . Please include the following items when submitting your revised manuscript:

We look forward to receiving your revised manuscript.

Kind regards,

Enzo Pasquale Scilingo, Ph.D.

Academic Editor

PLOS ONE

Journal Requirements:

Reviewers' comments:

Reviewer's Responses to Questions

**Comments to the Author**

1. If the authors have adequately addressed your comments raised in a previous round of review and you feel that this manuscript is now acceptable for publication, you may indicate that here to bypass the “Comments to the Author” section, enter your conflict of interest statement in the “Confidential to Editor” section, and submit your "Accept" recommendation.

Reviewer #1: All comments have been addressed

Reviewer #3: (No Response)

2. Is the manuscript technically sound, and do the data support the conclusions?

Reviewer #1: Yes

Reviewer #3: Yes

3. Has the statistical analysis been performed appropriately and rigorously? 

Reviewer #1: Yes

Reviewer #3: Yes

4. Have the authors made all data underlying the findings in their manuscript fully available?

Reviewer #1: No

Reviewer #3: Yes

5. Is the manuscript presented in an intelligible fashion and written in standard English?

Reviewer #1: Yes

Reviewer #3: Yes

6. Review Comments to the Author

Reviewer #1: Thank you for considering the points raised in the first review. I have just three very minor clarifications.

1. Line 151-154: “who have been registered with their general practice for at least one year; who, additionally, either have a record of a SNOMED-CT diagnostic code of interest within ten years prior to 31st December 2021 (see link associated with "Mental disorder | SCTID: 74732009 + child codes" in S2 Table), and who have a record of prescriptions for medicines of interest (Table 1) within ten years prior to 31st December 2021”

Should this be “either have a record of a SNOMED-CT diagnostic code… or who have a record…”, or if they needed both perhaps the word “either” needs removing?

2. Line 192: “Controls’ were those records that did not meet these two criteria.”

I presume this is anyone that didn’t meet either of these criteria? So someone with recent medication but no diagnosis becomes a control as does someone with a diagnosis in the absence of medication, and people that have neither? This could be a little clearer.

3. Thank you to the reviewers for providing a table 1 in the supplement. While I agree these tables may be mis/over interpreted they do provide useful context; for example for other researchers who may want to do similar in other regions. I am not sure your labels match your numbers for age in this table though, with e.g. max age listed as 30/40.

Reviewer #3: Thank you for the opportunity to review this interesting paper on the features of complex mental health difficulties in primary care records, which makes an important contribution to the literature. Key strengths of the study include the use of multiple feature types with several levels. Even though many of the patients with complex mental health difficulties remain ‘unseen’ following this methodology, the authors provide thoughtful reflections on the challenges of using electronic health record data to identify these patients. I have also read the reviewer reports and author responses from the first round of review, where the authors have addressed the reviewer feedback. I have minor points for the authors to address.

1. The antecedent feature set includes adverse childhood experiences; however, these experiences are not defined/specified in the text. Which adverse childhood experiences were included in the antecedent feature set? The authors cite a study in the discussion which found substantial under-coding of adverse childhood events, therefore we would expect high missingness in this feature set.

2. In the discussion, the authors refer to qualitative work undertaken as part of their wider work in this area and say that this was used to inform the quantitative study. From what I can see, this qualitative study is currently unpublished. The mixed-methodology should be made clearer - please add further information around how the findings from the qualitative study informed this study in the methods, for example if this informed the selection of features?

3. The article requires careful proof-reading for typos/phrasing issues, particularly:

o Line 55, comma needed after (dysthymia).

o Line 71, ‘,,,’ should be one comma, with a lower-case S for ‘similarly’.

o Line 137, remove ‘the’ before England.

o Line 187, removing ‘excluding those from our exclusion list’ as this phrase is redundant.

o Figure 1 title is quite long – may I suggest adding the sentences after ‘our records-based definition…’ as a footnote. Same for figs 2 and 3, after A). It seems a few of the figures have footnotes (e.g. Fig 5 includes a footnote entitled ‘10’, while others don’t). These figures are not titled/footnoted consistently, please can authors proof-read all figure titles/footnotes.

o Table 4, following ‘Active…’ should also be a footnote.

o Line 407-408, thirdly and fourthly repeated in same sentence.

7. PLOS authors have the option to publish the peer review history of their article (what does this mean? ). If published, this will include your full peer review and any attached files.

**Do you want your identity to be public for this peer review?** For information about this choice, including consent withdrawal, please see our Privacy Policy .

Reviewer #1: No

Reviewer #3: No

---

## [Author Response · Author response to Decision Letter 2]

26 Feb 2025

1. Line 151-154: “who have been registered with their general practice for at least one year; who, additionally, either have a record of a SNOMED-CT diagnostic code of interest within ten years prior to 31st December 2021 (see link associated with "Mental disorder | SCTID: 74732009 + child codes" in S2 Table), and who have a record of prescriptions for medicines of interest (Table 1) within ten years prior to 31st December 2021”

Should this be “either have a record of a SNOMED-CT diagnostic code… or who have a record…”, or if they needed both perhaps the word “either” needs removing?

Thank you for spotting the typo. The “either” is correct. I have changed the “and” to an “or”.

2. Line 192: “Controls’ were those records that did not meet these two criteria.”

I presume this is anyone that didn’t meet either of these criteria? So someone with recent medication but no diagnosis becomes a control as does someone with a diagnosis in the absence of medication, and people that have neither? This could be a little clearer.

Apologies for the ambiguity. ‘Controls’ were those who did not meet at least one criteria. We had a preference for specificity of cases over sensitivity. I have changed the wording to “‘Controls’ were those records that did not meet at least one of these criteria.”

3. Thank you to the reviewers for providing a table 1 in the supplement. While I agree these tables may be mis/over interpreted they do provide useful context; for example for other researchers who may want to do similar in other regions. I am not sure your labels match your numbers for age in this table though, with e.g. max age listed as 30/40.

I have identified the maximum age issue as a typo. I’ve corrected the order of labels in S3 Table.csv, which should have been {min, 1st quartile, median, 3rd quartile, max, arithmetic mean} instead of {min, max, median, Amean, quart1, quart3}. I have also corrected the labelling in the publicly-available Jupyter Notebook that creates the table (https://github.com/ConnectedBradford/CB_1759_Joining-Primary-and-Secondary-Care-/blob/main/code/UNSEEN_create_caseness_variable.ipynb).

1. The antecedent feature set includes adverse childhood experiences; however, these experiences are not defined/specified in the text. Which adverse childhood experiences were included in the antecedent feature set? The authors cite a study in the discussion which found substantial under-coding of adverse childhood events, therefore we would expect high missingness in this feature set.

Apologies for being unclear. I have replaced “e.g. adverse childhood experiences” with two examples of concepts that we used to represent adverse childhood experiences, “e.g. child abuse, abandonment, etc”. We did not define a feature for adverse childhood experiences, instead opting for multiple features indicating the variety of such experiences. Thus, I can’t check the missingness of the concept of adverse childhood experiences.

The full list of features used in the antecedent feature set are shown in the publicly-available Jupyter Notebook that details the Python code (https://github.com/ConnectedBradford/CB_1759_Joining-Primary-and-Secondary-Care-/blob/main/code/UNSEEN_create_feature_sets_appendix3.ipynb).

2. In the discussion, the authors refer to qualitative work undertaken as part of their wider work in this area and say that this was used to inform the quantitative study. From what I can see, this qualitative study is currently unpublished. The mixed-methodology should be made clearer - please add further information around how the findings from the qualitative study informed this study in the methods, for example if this informed the selection of features?

Apologies for the lack of clarity. I have now included a brief explanation within the ‘Features and feature sets’ section. In it, I explain a little more about how the interviews inspired our features, and I explain that the source of each feature is summarised in Table S3 where ‘fs_literature’ indicates a feature inspired by our literature review, ‘fs_interviews’ indicators a feature inspired by our interview study, and ‘fs_clinician’ indicators a feature inspired by the clinical members of the research team.

3. The article requires careful proof-reading for typos/phrasing issues, particularly:

o Line 55, comma needed after (dysthymia).

o Line 71, ‘,,,’ should be one comma, with a lower-case S for ‘similarly’.

o Line 137, remove ‘the’ before England.

o Line 187, removing ‘excluding those from our exclusion list’ as this phrase is redundant.

o Figure 1 title is quite long – may I suggest adding the sentences after ‘our records-based definition…’ as a footnote. Same for figs 2 and 3, after A). It seems a few of the figures have footnotes (e.g. Fig 5 includes a footnote entitled ‘10’, while others don’t). These figures are not titled/footnoted consistently, please can authors proof-read all figure titles/footnotes.

o Table 4, following ‘Active…’ should also be a footnote.

o Line 407-408, thirdly and fourthly repeated in same sentence.

I’m grateful for your keen eye. These typos have been corrected.

I respectfully disagree that “excluding those from our exclusion list” is a redundant phrase. Without it, the reader it not made aware that exclusions have been made.

Regarding figures and tables, I have moved the explanations to footnotes or to the main text, or removed the explanations entirely. I find it much messier and more difficult to digest the figures and tables, now; the reader has to jump around the document to piece together their understanding. The original titles might have been long but they presented all the information necessary to understand the table in one place. I request that the editor decide which style to use.

Figure 5 does not include a footnote entitled ‘10’; the ‘10’ is the base of the logarithm being referred to in the previous sentence.

---

## [Decision Letter · Decision Letter 2]

28 Mar 2025

Identifying primary-care features associated with complex mental health difficulties

PONE-D-24-01070R2

Dear Dr. McInerney,

We’re pleased to inform you that your manuscript has been judged scientifically suitable for publication and will be formally accepted for publication once it meets all outstanding technical requirements.

Kind regards,

Sreeram V. Ramagopalan

Academic Editor

PLOS ONE

Additional Editor Comments (optional):

Reviewers' comments:

Reviewer's Responses to Questions

**Comments to the Author**

1. If the authors have adequately addressed your comments raised in a previous round of review and you feel that this manuscript is now acceptable for publication, you may indicate that here to bypass the “Comments to the Author” section, enter your conflict of interest statement in the “Confidential to Editor” section, and submit your "Accept" recommendation.

Reviewer #1: All comments have been addressed

Reviewer #3: All comments have been addressed

2. Is the manuscript technically sound, and do the data support the conclusions?

Reviewer #1: Yes

Reviewer #3: Yes

3. Has the statistical analysis been performed appropriately and rigorously? 

Reviewer #1: Yes

Reviewer #3: Yes

4. Have the authors made all data underlying the findings in their manuscript fully available?

Reviewer #1: No

Reviewer #3: No

5. Is the manuscript presented in an intelligible fashion and written in standard English?

Reviewer #1: Yes

Reviewer #3: Yes

6. Review Comments to the Author

Reviewer #1: (No Response)

Reviewer #3: The authors have addressed my comments and I recommend this paper for publication. The author has requested for the editor to make a final decision on the formatting of the figures.

7. PLOS authors have the option to publish the peer review history of their article (what does this mean? ). If published, this will include your full peer review and any attached files.

**Do you want your identity to be public for this peer review?** For information about this choice, including consent withdrawal, please see our Privacy Policy .

Reviewer #1: No

Reviewer #3: No

---

## [Editor Report · Acceptance letter]

PONE-D-24-01070R2

PLOS ONE

Dear Dr. McInerney,

I'm pleased to inform you that your manuscript has been deemed suitable for publication in PLOS ONE. Congratulations! Your manuscript is now being handed over to our production team.

Kind regards,

on behalf of

Dr. Sreeram V. Ramagopalan

Academic Editor

PLOS ONE